# Examining the Feasibility of an Application-Based Patient-Reported Outcome Monitoring for Breast Cancer Patients: A Pretest for the PRO B Study

**DOI:** 10.3390/ijerph19148284

**Published:** 2022-07-07

**Authors:** Anna Maria Hage, Pimrapat Gebert, Friedrich Kühn, Therese Pross, Ulrike Grittner, Maria Margarete Karsten

**Affiliations:** 1Department of Gynecology and Breast Center, Charité—Universitätsmedizin Berlin, Charitéplatz 1, 10117 Berlin, Germany; anna-maria.hage@charite.de (A.M.H.); fiete.kuehn@gmx.de (F.K.); therese.pross@charite.de (T.P.); 2Berlin Institute of Health at Charité—Universitätsmedizin Berlin, Charitéplatz 1, 10117 Berlin, Germany; pimrapat.gebert@charite.de (P.G.); ulrike.grittner@charite.de (U.G.); 3Institute of Biometry and Clinical Epidemiology, Charité—Universitätsmedizin Berlin, Charitéplatz 1, 10117 Berlin, Germany

**Keywords:** patient reported outcomes, metastatic breast cancer, health apps, personalized medicine, ePROs, mobile health application, mHealth

## Abstract

In preparation for the PRO B study which aims to examine the effects of an app-based intensified patient-reported outcome (PRO) monitoring for metastatic breast cancer patients, prior assessment of its feasibility was carried out. Sixteen breast cancer patients visiting the breast cancer unit at Charité were recruited and downloaded an app connected to an ePRO system. They received electronic questionnaires on two occasions (baseline and the following week) and were subsequently contacted for a semi-structured phone interview for evaluation. Eleven participants answered at least one questionnaire. Some participants did not receive any or only a part of the questionnaires due to technical problems with the app. Participants who completed the evaluation questionnaire (*n* = 6) were overall satisfied with the weekly PRO questionnaire. All interviewed (*n* = 11) participants thought it was feasible to answer the PRO questionnaires on a weekly basis for one year, as planned in the PRO B study. The pretest revealed a need for major technical adjustments to the app because push notifications about the receipt of new questionnaires were not displayed on some smartphone models. Due to the low number of participants, generalization of the findings is limited to our specific context and study. Nevertheless, we could conclude that if technical aspects of the app were improved, the PRO B study could be implemented as planned. The ePRO questionnaire was considered feasible and adequate from the patients’ perspectives.

## 1. Introduction

The use of electronic patient-reported outcome (ePRO) measures in the care of oncologic patients is being increasingly studied. ePROs can collect real-time data about the patients’ general condition using digital surveys and, therefore, enable a fast and personalized monitoring strategy for various domains, such as symptoms, side effects of therapies, functioning, and/or quality of life (QoL). As physicians often underestimate the severity of symptoms and side effects [1,2], collecting information directly from patients might improve the accuracy of this subjective data and potentially improve the efficiency of clinical care [3]. Digital tools such as apps facilitate this process [4,5].

Patients with metastatic cancer are often affected physically and mentally because of the diagnosis itself, its treatment, or coexisting conditions [6]. Basch et al., used an intensified digital PRO-monitoring on 766 cancer patients under chemotherapy, and detected clinical benefits such as improved QoL and a reduction in unplanned hospitalization and ER visits [7], as well as a statistically significant increase in overall survival [8]. Another study by Denis et al. also examined an electronic PRO-monitoring with alerts in advanced-staged lung cancer patients, and found improvements in overall survival, as well as better performance status at the first relapse compared to standard care [9,10].

While designing the PRO B study [11] which aims to evaluate the effects of an app-based PRO-monitoring with alerts on metastatic breast cancer patients’ fatigue levels and QoL, we modified an existing digital symptom monitoring tool specifically for the project. The adapted system measures ePROs weekly using an app, then graphically displays the results for clinicians in a web tool and generates alerts in the case of deteriorating ePROs. The alert notifies the treating physician to get in contact with the patient to determine if further interventions are necessary.

In preparation for the PRO B study, prior testing of the ePRO system was conducted and evaluated by breast cancer patients. The study team wanted to include patients’ perspectives on the planned PRO B study, since patient-centered care is a critical component of care quality.

Although prior testing of such systems is often recommended [12,13], usability evaluation of eHealth apps has been an under-represented topic in publications [14].

## 2. Materials and Methods

### 2.1. Aim of the Study

The pretest aimed to explore the feasibility of the planned PRO B study. Therefore, the main areas of interest were the patient’s experience with the app used in the study, the patient’s experience with the PRO B study questionnaires, and the feasibility of the PRO B study from the patient’s perspective.

Insights and conclusions from the pretest will serve as suggestions for adjustments before starting the main phase of the PRO B study.

### 2.2. Study Design

This pretest was a single-arm pilot study and took place at the Breast Center of Charité—Universitaetsmedizin Berlin between February and March 2021. The study was approved by the Charité Ethics board (No. EA1/318/20). Breast cancer patients were invited to participate. They were instructed to download the PRO application “*PatientConcept*” on their smartphone or tablet after giving written informed consent. The app then automatically generated an ID, which served as a pseudonym in the study and was communicated to the study team so that no identifying personal data was included in the ePRO system.

The ePRO system assigned questionnaires to the IDs in conjunction with push notifications on the patients’ smartphones. Questionnaires were available for 48 h and disappeared after the 48 h period. The survey was performed at baseline and one week later. At baseline, anamnestic, socio-demographic, and PRO questionnaires were sent from the ePRO system to the app. A week later, patients were asked to complete a second PRO assessment and an evaluation questionnaire inquiring about the pretest.

Afterwards, we also conducted semi-structured telephone interviews with the participants to further assess their feedback on the pretest to explore additional information that might not have been represented in the evaluation questionnaire (Figure 1).

### 2.3. Participants

Female breast cancer patients of legal age were eligible for participation. Moreover, the participants were required to have access to the internet via a smartphone or tablet and be able to consent to participation. Patients visiting the breast cancer unit at Charité were invited to participate. However, unlike the PRO B study, inclusion in the pretest included, not only metastatic breast cancer patients, but also, patients with early breast cancer.

### 2.4. Study Instruments

***Baseline questionnaires*** collected data on medical history and socio-demographics [15].

***PRO questionnaires*** for the PRO B study were compiled from items of the EORTC (European Organisation of Research and Treatment of Cancer) CAT (computerized adaptive testing) item bank [16]. Depending on the week, the number of questions asked ranged from 51 to 54 items concerning the domains QoL, physical, role, cognitive, emotional, and social functioning as well as the symptom scales of fatigue, pain, nausea/vomiting, dyspnea, insomnia, appetite loss, constipation, diarrhea, and financial difficulties. In the pretest, the questionnaires were assigned twice on a week interval. 

***The pretest evaluation questionnaire*** consisted of two parts: opinion about the weekly PRO questionnaire and opinion about the smartphone application. Both parts used closed questions with a Likert-scale and a few open questions (Appendix A). The first part was developed by the study team and consists of five questions with a 5-Likert scales. Internal consistency using Cronbach’s alpha was 0.77 (acceptable). The second part was partly derived from the mHealth App Usability Questionnaire (MAUQ) [17] with a 7-point Likert-scale: 1 = strongly disagree to 7 = strongly agree) and the Cronbach’s alpha was 0.87 (good internal consistency). Technical problems were assessed using open questions with a free text response option.

***A semi-structured telephone interview guide*** was developed in a multi-stage consensus process by the study team, which included a senior breast surgeon/gynecologist, biostatisticians, a study nurse, and a psychologist pursuing a deductive and inductive strategy. Questions were developed attempting to verify hypotheses generated from the study teams’ clinical expertise as well as being explorative in nature in order to freely capture the patients’ perspective and gather information in addition to the quantitative assessment (Appendix A). The aim was to identify additional challenges that might not have been represented with the evaluation questionnaire.

### 2.5. Semi-Structured Interviews

The telephone interviews were conducted by two physicians (A.M. and F.K.) from the PRO B team using the developed guide. The physicians were not involved in the clinical care of the participants.

Participants were then interviewed individually by phone in German language. The interview questions are presented in the Appendix A.

Since the interviews were not recorded and were only documented in the form of notes by the interviewing physicians, transcriptions were not available and, therefore, analysis was limited to a descriptive summary of the results.

### 2.6. Analysis

Exploratory statistics, such as median, minimum, maximum, and absolute and relative frequencies were used to describe quantitative data using Stata IC15 (StataCorp, 2017, College Station, TX, USA).

A biostatistician sorted the notes from the interviews into content domains (perception of workload and duration, perception of the PRO questionnaires, perception of using the application, general suggestions, and using the application in the future and recommendation) and summarized the results using frequency counts.

## 3. Results

Sixteen female patients were recruited. Four did not answer any questionnaires and were, therefore, excluded. Eleven patients answered questionnaires in the app at least once, and one patient received the questionnaires via e-mail due to technical problems. Mean age was 47 years (SD 11, range 31–70 years), two participants were diagnosed with metastatic disease, while 10 were diagnosed with early breast cancer.

Only six of the twelve participants answered the evaluation questionnaire at the end of the pretest.

Due to major technical problems, the app, in some cases, did not display notifications on the mobile phone, leaving the participants unaware that there were new questionnaires available, and the questionnaires expired unanswered after 48 h.

Nevertheless, we were able to include eleven patients in the semi-structured interviews to further assess the experience with the ePRO tool.

### 3.1. Results of the Evaluation Questionnaire

#### 3.1.1. Patients’ Experiences with the PRO B Study Questionnaires

Five out of six patients who answered the evaluation questionnaire were satisfied or very satisfied with the PRO questionnaires and the time spent on them. Four of the patients reported that the questions were very easy to understand and that they were able to concentrate the entire time. However, five patients reported that the questions were only moderately relevant to them (Table 1).

Two patients reported that some questions seemed very similar, e.g., regarding fatigue and physical functioning. Three patients answered that the survey lacked important aspects of chemotherapy-related side effects, e.g., polyneuropathy. One patient wished for more questions regarding psycho-oncology, e.g., on worry and anxiety.

#### 3.1.2. Patients’ Experiences with the Application

The results show that patients who were able to answer the questionnaires were overall very satisfied with using the app (median score of 7 points (7 = highest score)) (Table 2).

### 3.2. Results of the Telephone Interview

The study team reached fourteen of the sixteen participants for the semi-structured telephone interview. Three of the participants did not participate in the telephone survey because they either did not receive any questionnaires (*n* = 2) or felt too ill to fill them out (*n* = 1). Overall, eleven patients were interviewed, of whom ten received and answered PROs at least once, and one participant reviewed the questionnaires after receiving them via e-mail instead of in the app (due to the technical problems).

Perception of workload and duration

Out of eleven participants, all considered it feasible to answer all items once a week for a year as planned in the intervention group of the PRO B study. Most of the patients completed the questionnaires in around 5–20 min.

Perception of the PRO questionnaires 

After being asked if the PRO-questionnaires covered the most relevant aspects, patients responded positively, seven patients gave minor suggestions about what they would like to additionally see represented in the PRO-questionnaire. Side effects of treatment were mentioned most frequently (*n* = 4), and digestion, mental health, sexual and social aspects were also mentioned to better represent the patients’ burdens. Four participants mentioned they felt that some questions seemed redundant and sounded too general. A participant remarked that the amount of housework varies for everyone and is, therefore, not a comparable term. One patient had a concern about emotional and financial problems related to the corona-virus pandemic, as it could have aggravated these problems. Another patient reported that it could be difficult to complete the quality-of-life questionnaire when a private event occurs that is not related to breast cancer. Moreover, one patient said that she was confused by the used ranking scale and remarked that she would feel more familiar with German school grades (1–6). However, all of the patients thought that the questions were helpful for the doctors to assess the patient’s health status.

Perception of using the application

Multiple patients reported having technical problems using the application, especially with getting push notifications. Therefore, they were left unaware of receiving questionnaires in the app. They had to check the app for the questionnaires frequently by themselves, otherwise the questionnaire would expire. No patient mentioned any problems in the app while answering the questionnaires.

General suggestions from participants

All of the patients considered the system useful. One would have preferred a different color scheme, as the color blue seemed too sterile and reminded her of a hospital. A more personalized greeting would have been nice. Another patient would have liked to receive an additional notification on the second day to be reminded about the availability of questionnaires. It was also suggested to add a short explanation or instruction about the questions concerning health status. One patient would have liked to see the results of their health state assessment graphically displayed in the app and, if possible, receive a weekly report. She also suggested including a short explanation and information on each question in the app. Moreover, the patient would have liked to receive some information on the latest research in the app.

Using the application in the future and recommending it to others

All but one of the patients interviewed (*n* = 10) would like to use the system and concept in the future and would recommend it to other patients. One patient reported that she already felt adequately treated and, therefore, would not need to use the application further in the future.

Suggestions for motivating the control group

The control group in the PRO B study will answer questionnaires only quarterly, and alerts will not be generated in the case of deteriorating PROs [9]. The treating physicians will not have access to their answers during the study but will receive an overview at the end of the trial. As there is no primary benefit to answering the questionnaires for the control group, we wanted to examine if we should expect a high drop-out or non-compliance rate. Therefore, we asked the participants for suggestions on how to motivate the patients in the control group. Two participants mentioned that the control group could be questioned even more frequently. However, most of them suggested using a reminder notification and felt that the opportunity to contribute to research was incentive enough. For high compliance, the study team should explain the potential advantages of the project for future patients afflicted with metastatic breast cancer.

## 4. Discussion

In summary, the results show positive experiences in both the evaluation questionnaire and the telephone interview among participants who used the PRO B tool. Since interviews were not recorded and transcribed, the analysis is limited to descriptive summaries. Nevertheless, we could collect valuable insights from patients’ perspectives and confirm the overall acceptability of the questionnaire and the PRO B study design. Participants were generally satisfied and found that the app could be useful for tracking their HRQoL and symptoms. The study team was previously concerned that the time spent on answering the questionnaires might be overwhelming. However, the pretest allayed those concerns. From the patients’ points of view, answering the questionnaire on a weekly basis in the upcoming PRO B study is a tolerable burden that takes up a reasonable amount of time. Accordingly, after finding solutions to the severe technical problems, the app is feasible and possible to use for monitoring PROs in breast cancer patients. Moreover, downloading the *PatientConcept* application is free of charge and has been used to support communication between physicians and patients [18,19].

The pretest revealed a need for technical improvement to the ePRO system and app. Therefore, a function documenting the delivery of the questionnaires to the participants was established after the pretest. Most participants did not answer the full set of questionnaires during the pretest, mostly because they did not receive any or were not aware of them because of missing push-notifications. Receipt of push-notifications was improved in a subsequent app-update.

To ensure compliance in the upcoming PRO B study, non-responding participants will receive push notifications as a reminder after 24 h.

The limitations of this pretest are the small number of patients and short study period. Due to a longer than expected time for implementation of the ePRO system and the upcoming start of the PRO B study, the duration of the pretest was restricted. Also, most of the participants did not suffer from metastatic breast cancer—unlike the inclusion criteria of the PRO B study. Therefore, the patients’ points of view and needs might differ.

Due to major technical problems with the app, we cannot draw conclusions about the response rate. Also, there was no testing of the alarm function in case of worsening scores because of the short study period. Moreover, qualitative assessment was not possible since the interviews were not recorded and transcribed. Nevertheless, we were still able to get an impression of the patients’ views, and we consider their suggestions valuable.

## 5. Conclusions

As every medical procedure should primarily focus on the patient on whom it is performed, the inclusion of the patient’s perspective in the development of new forms of care such as ePRO systems is crucial.

Unfortunately, in this pretest, findings from this study are limited to our specific study and context due to the low number of patients and the major unexpected technical problems with the app. Nevertheless, we note that none of the participants raised major concerns regarding the time spent answering the questionnaires via app or the content of the questionnaires. Therefore, we conclude that the PRO B study design and questionnaires are feasible and adequate to implement as planned. Major technical issues regarding push notifications in the app were identified and rectified after the pretest, but considerably delayed the start of enrollment for the PRO B study. When including new technologies in clinical studies, technical difficulties cannot be underestimated. Therefore, calculating a sufficient amount of time from pretest to enrollment is crucial.

## Figures and Tables

**Figure 1 ijerph-19-08284-f001:**
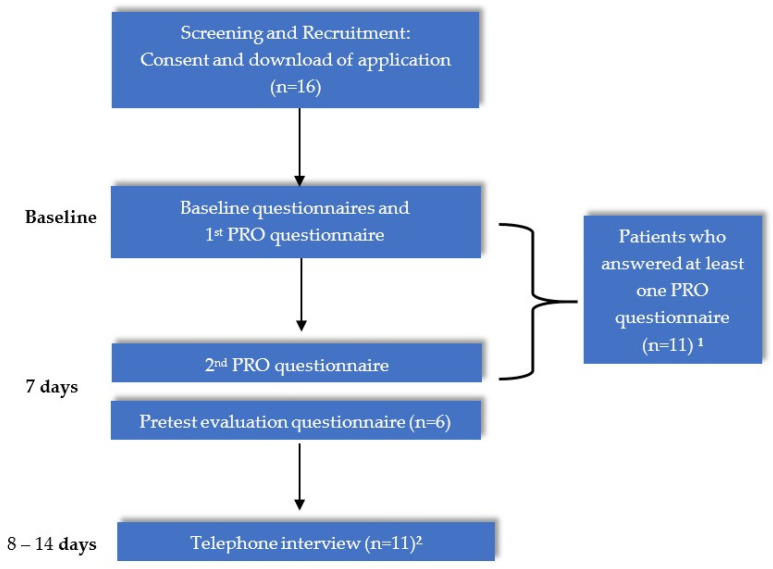
The pretest study flow. ^1^ Six out of eleven patients answered PRO questionnaires on both occasions; ^2^ The interview includes one patient who reviewed the questionnaires via e-mail and did not answer in the app.

**Table 1 ijerph-19-08284-t001:** General opinion about PRO B questionnaires in the application (*n* = 6).

General Opinion about PRO B Questionnaires	Very Much	Much	Moderate	A Little	Not at All
Overall satisfaction	3	2	1	0	0
The questions are understandable	4	1	1	0	0
The questions are relevant for breast cancer patients	0	1	5	0	0
I am able to concentrate while answering the questionnaires	4	1	1	0	0
Time needed to answer the questionnaires is appropriate	2	3	1	0	0

**Table 2 ijerph-19-08284-t002:** Opinion on using the application (Likert scale ranges from 1 = totally disagree to 7 = totally agree) (*n* = 6).

Aspects	Questions	Median (Min, Max)
Ease of use and satisfaction	The app was easy to use	7 (5, 7)
	It was easy for me to learn to use the app	7 (7, 7)
	I like the interface of the app	6 (4, 7)
	The information in the app was well organized, so I could easily find the information I needed	6 (4, 7)
	I feel comfortable using this app in social settings	6 (4, 7)
	The amount of time involved in using this app has been fitting for me	7 (5, 7)
	I would use this app again	7 (4, 7)
	This app offers great opportunity to improve care for breast cancer patients	6 (4, 7)
	Overall, I am satisfied with this app	7 (5, 7)
System information arrangement	Whenever I made a mistake using the app, I could correct it easily and quickly	6.5 (1, 7)
	The navigation was simple and clearly structured	7 (5, 7)

## Data Availability

The data analyzed during this study are not permitted to be made publicly available.

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
