# Peer review of "Examining the Feasibility of an Application-Based Patient-Reported Outcome Monitoring for Breast Cancer Patients: A Pretest for the PRO B Study"

_ijerph, 2022, doi:10.3390/ijerph19148284_

Round 1
Reviewer 1 Report
Please see the attached Word document for a comprehensive review. Thank you.

Reviewer 2 Report
This is an interesting feasibility analysis of the pretest of the PRO B study based on semi-structured telephone interviews. Due to the small number of participants in the pretest and the high number of drop-outs due to technical issues the informative value is limited. Nevertheless the manuscript provides important information for the planning and conception of PRO monitoring studies.
Reviewer 3 Report
Interesting study of actual relevance . However, the manuscript needs some improvements :
1.- Aim of the study. Please clarify the main aim. Pre-test of the acceptation, use, potential etc of an app ? other ?". descriptive study, experimental study. other ?
It is today, not possible to read or understand how the analysis of "the technical functions of the adapted ePRO system" is performed. No approach to perform the study is presented. In addition to this, there are no data related to "technical issues".
2.- The method chapter must be improved.
Nothing is said about the age of the participants or how they were selected.
Which approach support the questionnaire used ? .
The consistency of the questions with the aim " technical functions of a system" is unclear.
The questions are related to other issues such as usability, acceptability, adaption, adoption etc.
The number of respondents is low.
It seems that this is a descriptive study difficult to see that an statistical analysis has been performed. The study describe the answers from 16 respondents.
If the results cannot be generalized, as the authors pointed out, what is needed to replicate the study ?
An analysis of the consistency of the study with some approach, and some comments about the accuracy of the data collected is needed.
Some healthcare professionals were interviewed. What are the conclusions from the interviews?
It is possible to compare the answers from the patients with the answers from the healthcare professionals?
It is correct to call a professional group as "control group" when the study focus on patients experiences, acceptation of an app. etc? . The two groups have different expectations and experiences. Have the authors consider that this can influence the accuracy of the results ?
How has the answers been validated ?
3.- Is the study consistent with previous studies ?
Author Response
Thank you very much for your reveiw.
Please see the attachment.

Reviewer 4 Report
This is a well performed and presented study, which, although the patients`collective was relatively small, is of high clinical relevance. Thus, it is suggested to be accepted in the present form.
Author Response

(The authors gave the same response as above.)

Round 2
Reviewer 1 Report
Dear authors,
Thank you for letting me read your revised version and your response. The manuscript has significantly improved in quality. The aim of the study is clearer.
General comments:
The paper could further be improved in quality if you could state your conclusions clearer as an answer to your feasibility study. The conclusions you have drawn can be summarised into two major points:
- Technical aspects of the app need to be improved
- The questionnaire is considered feasible and adequate from patients’ perspectives, thus the overall PROB study design, including the questionnaire, does not have to be changed
I suggest you clearly state these points in the Abstract and the Conclusion parts.
Abstract:
Line 24-26, suggestion: "Due to the low number of participants, generalization of the findings is limited to our specific context and study. Nevertheless, we could conclude that if technical aspects are improved, the PRO B Study can be implemented as planned. The ePRO questionnaire was considered feasible and adequate from patients´ perspectives.
Introduction
This has improved with relevant literature added.
Line 55, suggestion: “the study team wanted to include patients´ perspectives…...”
Line 58-60, suggestion: “Although prior testing of such systems is often recommended [12,13], usability evaluation of eHealth apps has been an underrepresented topic in publications [14].
Aim is clearer now
Line 67: “will serve as”
Discussions
Line 258, suggestion: “Since interviews were not not recorded and transcribed, the analysis is limited to descriptive summaries. Nevertheless, we could collect valuable insights from patients´ perspectives and confirm the overall acceptability of the questionnaire and the PRO B study design.”
Conclusions
I suggest to rephrase the below sentence. It is stating the obvious and too vague. Make it more specific to ePROs, digital PRO monitoring.
“As every medical procedure should primarily focus on the patient it is performed on, the inclusion of the patient perspective in the development of new forms of care is crucial.”
Moderate English changes are required
“Conclusions are limited” is not a good English expression – see also my suggestion in Abstract section.
Either
- “findings from this study are limited to the small group of participants who took part in the study” or
- the “generalisation of the conclusions drawn are limited to our specific study and context”
“time expenditure” is not an adequate expression – “time spent on the app”.
As mentioned in the Abstract section, I will make the conclusion clearer and stronger – something like…
- “we conclude that if technical aspects are improved, the PRO B Study can be implemented as planned as the ePRO questionnaire was considered feasible and adequate from patients´ perspectives”.
With best regards,
(Attached texts in WORD doc are the same as above).

Author Response
Dear Reviewer,
First, we thank you and appreciate your taking the time to review our revised version.
Your comments and suggestions helped us and improved our manuscript considerably.
We have revised our manuscript and responded to your comments point by point, as you can see in the track-change version attached.
General comments:
The paper could further be improved in quality if you could state your conclusions clearer as an answer to your feasibility study. The conclusions you have drawn can be summarized into two major points:
- Technical aspects of the app need to be improved
- The questionnaire is considered feasible and adequate from patients’ perspectives, thus the overall PROB study design, including the questionnaire, does not have to be changed
I suggest you clearly state these points in the Abstract and the Conclusion parts.
- Thank you for your suggestions and the summary. That makes our paper considerably clearer. We have added these two points in the abstract (line 25-30) and the conclusion (line 297-302).
Abstract:
Line 24-26, suggestion: "Due to the low number of participants, generalization of the findings is limited to our specific context and study. Nevertheless, we could conclude that if technical aspects are improved, the PRO B Study can be implemented as planned. The ePRO questionnaire was considered feasible and adequate from patients´ perspectives.
- Thank you for your suggestion. We did the change in the abstract (line 25-30).
Introduction
This has improved with relevant literature added.
Line 55, suggestion: “the study team wanted to include patients´ perspectives…...”
- We did the change in line 61-62.
Line 58-60, suggestion: “Although prior testing of such systems is often recommended [12,13], usability evaluation of eHealth apps has been an underrepresented topic in publications [14].
- We did the change in line 64-67.
Aim is clearer now
Line 67: “will serve as”
- Thank you. We did the change in line 76.
Discussions
Line 258, suggestion: “Since interviews were not recorded and transcribed, the analysis is limited to descriptive summaries. Nevertheless, we could collect valuable insights from patients´ perspectives and confirm the overall acceptability of the questionnaire and the PRO B study design.”
- We revised the sentence as to your suggestion in the discussion line 259-262.
Conclusions
I suggest to rephrase the below sentence. It is stating the obvious and too vague. Make it more specific to ePROs, digital PRO monitoring.
“As every medical procedure should primarily focus on the patient it is performed on, the inclusion of the patient perspective in the development of new forms of care is crucial.”
- We revised the sentence in line 292-294.
“As every medical procedure should primarily focus on the patient it is performed on, the inclusion of the patient perspective in the development of new forms of care like ePRO systems is crucial.”
Moderate English changes are required
“Conclusions are limited” is not a good English expression – see also my suggestion in Abstract section.
Either
- “findings from this study are limited to the small group of participants who took part in the study” or
- the “generalisation of the conclusions drawn are limited to our specific study and context”
- We revised the sentence in the conclusion line 295-302.
“time expenditure” is not an adequate expression – “time spent on the app”.
- We changed “time expenditure” to “time spent answering the questionnaires via the app” (line 299)
Best regards,
All authors
Reviewer 3 Report
NO further comments. The authors have reviewed the manuscript and answered comments from the reviewers
Author Response
Dear Reviewer,
We thank you and appreciate your taking the time to review our revised version.
Best regards,
All authors